# Comparison of thyroid hormones reference intervals based on thyroid antibody levels: A multicenter study

Havva Beyter[1], Osman Acar[2], Oytun Portakal[3], Özlem Gülbahar[4], Gülsüm F. Türkeş[5], Neslihan Yıldırım Saral[6], Muhittin Abdülkadir Serdar[7,8]*, Aysel Ozpinar[7,8]*

**1** Department of Biochemistry and Molecular Biology, Graduate School of Science, Acibadem Mehmet Ali Aydinlar University, Istanbul, Türkiye, **2** Department of Medical Biochemistry, School of Medicine, Acibadem Mehmet Ali Aydinlar University, Istanbul, Türkiye, **3** Department of Medical Biochemistry, School of Medicine, Hacettepe University, Istanbul, Türkiye, **4** Department of Medical Biochemistry, School of Medicine, Gazi University, Istanbul, Türkiye, **5** Department of Medical Biochemistry, School of Medicine, Ankara University, Istanbul, Türkiye, **6** Acibadem Labmed Clinical Laboratories, Department of Medical Biochemistry, School of Medicine, Acibadem Mehmet Ali Aydinlar University, Istanbul, Türkiye, **7** Department of Medical Biochemistry, School of Medicine, Acibadem Mehmet Ali Aydinlar University, Istanbul, Türkiye, **8** Department of Biochemistry and Molecular Biology, Graduate School of Health Sciences, Acibadem Mehmet Ali Aydinlar University, Istanbul, Türkiye

* muhittin.serdar@acibadem.edu.tr (MAS), Aysel.Ozpinar@acibadem.edu.tr (AO)

## Abstract

### Objectives

This study aimed to determine reference intervals (RI) for thyroid hormones based on thyroid antibody levels using different autoanalyzer kits.

### Methods

RI for Thyroid-stimulating hormone (TSH), free thyroxine (fT$_4$) and free triiodothyronine (fT$_3$) were determined according to thyroid antibody levels and independently of thyroid antibody levels using the R statistical program and RefineR algorithm.

### Results

Significant differences in RIs were found between antibody-positive (Ab(+)) and antibody-negative (Ab(–)) individuals. TSH RI varied most notably in Abbott and Siemens analyzers. In females, Abbott showed higher TSH RIs in the Ab(+) group (0.41–7.44 mU/L) than in Ab(–) (0.24–3.50 mU/L). In males, Roche and Beckman exhibited the greatest differences (Roche Ab(+): 0.19–5.77; Ab(–): 0.44–3.63; Beckman Ab(+): 0.12–5.23; Ab(–): 0.39–3.96 mU/L). For fT4, Roche showed increased RIs in females with Ab(+) status (11.42–20.42 vs. 10.34–19.35 pmol/L). In males, Beckman and Siemens autoanalyzers also indicated notable differences.

**Data availability statement:** All relevant data are within the manuscript and its Supporting Information files.

**Funding:** The author(s) received no specific funding for this work.

**Competing interests:** The authors have declared that no competing interests exist.

## Conclusion

Antibody status significantly affects thyroid hormone RI, particularly for TSH. These findings highlight the need for antibody-specific RI and further standardization in establishing reference intervals.

## Introduction

The thyroid gland is a key endocrine organ that synthesizes essential hormones such as triiodothyronine (T3) and thyroxine (T4), playing a crucial role in regulating metabolism. These hormones influence various biological processes, including lipid metabolism, cardiovascular function, bone health, and cognitive performance, necessitating an accurate assessment of thyroid function [1,2]. Thyroid-stimulating hormone (TSH), secreted by the pituitary gland, regulates thyroid hormone production and is considered one of the most important biomarkers of thyroid function. In addition to TSH, T3, T4, thyroglobulin antibodies (Tg-Ab), and thyroid peroxidase antibodies (TPO-Ab) are essential screening tests for evaluating thyroid function [3,4].

Thyroid disorders are among the most prevalent endocrine disorders worldwide, with their incidence increasing with age. Various physiological and pathological factors can influence thyroid hormone levels, including age, sex, pregnancy, acute or chronic illnesses, dietary iodine content, seasonal variations, and medication use [5–7].

Thyroid disorders are often associated with autoimmune diseases such as Hashimoto's thyroiditis and Graves' disease, which are characterized by the presence of TPO-Ab and Tg-Ab autoantibodies [8]. Although thyroid antibody positivity (Ab+) is an indicator of autoimmune thyroid disease, individuals with positive antibodies are not directly classified as patients nor subjected to immediate treatment [9]. Currently, treatment decisions are primarily based on TSH levels rather than antibody status [8].

Autoimmune thyroid diseases develop gradually over time. In the early stages, only antibody positivity is detected, whereas in later phases, thyroid function deteriorates, leading to significant changes in TSH levels [10,11]. In this process, the accuracy of RI used for thyroid function assessment becomes crucial. However, current RIs are often established without considering the presence of thyroid antibodies (Ab), which serve as early biochemical markers of thyroid autoimmunity. Most laboratories use refeence intervals for thyroid function evaluation without accounting for antibody status. Yet, antibody levels, as early indicators of autoimmune processes, have a significant impact on thyroid function.

In the current literature, it is observed that the RI for thyroid function is often determined without considering antibody status. Many studies establish these intervals by excluding antibody-negative individuals. To better represent a healthy population, we propose that reference intervals should be defined based on antibody-negative (Ab−) individuals. Antibody negativity (Ab−) is a critical criterion, and including these individuals in analyses will provide more accurate and clinically meaningful reference ranges. In this context, a healthy population should be defined as those who are antibody-negative. When determining reference intervals, TSH levels in

antibody-negative (Ab−) healthy individuals should be measured and used as the reference range to enhance the accuracy of thyroid function assessments.

In our study, we aim to emphasize the importance of defining reference intervals for thyroid function tests based on thyroid antibody status. Our hypothesis is that the reference intervals for antibody-positive (Ab+) individuals differ from those for antibody-negative (Ab−) individuals, and this difference may have significant implications for clinical diagnosis and treatment. We argue that thyroid function test reference intervals should be re-evaluated considering thyroid antibody status to ensure accurate interpretation of test results.

Moreover, reference intervals for thyroid function tests may vary not only based on antibody status but also due to demographic factors (such as ethnicity, age, and sex), body mass index, specific medication use, iodine status, and methodological differences [11–14,15,16]. Organizations such as the American Thyroid Association (ATA), the International Federation of Clinical Chemistry (IFCC), and the Clinical and Laboratory Standards Institute (CLSI) recommend that each laboratory establish population-specific reference intervals [12,17,18]. However, many laboratories and hospitals continue to use manufacturer-provided reference intervals, which are typically based on American and European populations. Previous studies have demonstrated significant ethnic differences in thyroid hormone reference intervals, emphasizing the need for population-specific ranges [11,15]. Therefore, RIs used for evaluating thyroid function should be tailored to the characteristics of the studied population [19,20].

In this study, we aimed to establish reference intervals for thyroid hormones in the Turkish population aged 18–50 using four different autoanalyzers (Architect i2000sr, Atellica IM, Modular E170, and DxI 800 Unicel). We evaluated the impact of thyroid antibody levels on these reference intervals and analyzed measurement differences between autoanalyzers. Our findings highlight the importance of considering antibody status and demographic factors when defining reference intervals for thyroid function tests.

## Materials and methods

### Study participants

This study was conducted as a retrospective analysis based on data obtained from laboratory records over a long period 01/01/2016 and 31/12/2023. Data were accessed for research purposes on 01/02/2024. The thyroid function test results collected from different centers were analyzed using data obtained from the laboratory records of these hospitals. The study was carried out as a multicenter study across different regions of Turkey, and the included centers represent the country's general demographic structure, including cosmopolitan cities such as Ankara and Istanbul. Thyroid hormone and thyroid antibody levels were measured using four different autoanalyzers: the Architect i2000sr (Abbott Laboratories, Abbott Park, Illinois, U.S.A), Atellica IM (Siemens Diagnostics, Tarrytown, NY), Modular E170 Analyzer (Roche Diagnostics, Germany), and DxI 800 Unicel (Beckman Coulter, USA). A total of 44,671 individuals, obtained from measurements performed in different laboratories, were included in this study. Of these individuals, 8,818 were male and 35,851 were female.

### Laboratory differences and measurement methods

In this study, thyroid parameters were measured using various autoanalyzers and immunoassay methods across different medical centers. Depending on the hospital, some Acıbadem hospitals used Roche analyzers, while others used Siemens analyzers. At Acıbadem Labmed Clinical Laboratory, thyroid parameters were measured using the Siemens Healthineers Atellica IM Analyzer with the chemiluminescent immunoassay (CLIA) method. Data from a total of 22,918 individuals (4,611 males and 18,307 females) collected between 2013 and 2023 were included in the analysis. During the same period, measurements were also performed at the same laboratory using the Roche Diagnostics Modular E170 Analyzer with the electrochemiluminescent immunoassay (ECLIA) method, and data from 10,334 individuals (1,915 males and 8,417 females) were included in the study. Furthermore, thyroid parameters were measured using the Architect i2000sr Analyzer with the CLIA method at Ankara Keçiören Training and Research Hospital. A total of 1,591 individuals (303 males

and 1,288 females) participated in this group. In addition, at Ankara Hacettepe University Faculty of Medicine Hospital, thyroid parameters were measured using the Beckman Coulter Access DxI 800 Unicel device with the CLIA method, including 1,989 males and 7,839 females, totaling 9,828 participants.

## Study design

In this study, cut off for thyroid antibody levels were determined based on the values provided in the kit inserts of four different autoanalyzers. The autoanalyzers and their manufacturer-reported reference values are as follows:

   Abbott; TG: 4.11 U/mL, TPO: 5.61 U/mL,
   Beckman; TG: <1 IU/mL, TPO: <10 IU/mL
   Roche; TG: 115 IU/mL, TPO: 34 IU/mL
   Siemens; TG: >4.5 IU/mL, TPO: >60 U/mL
   In Turkey, all laboratory reference intervals undergo a verification process before being implemented in clinical practice. Therefore, the manufacturer-provided RI were validated and used in this study. Thyroid antibodies were classified as positive or negative according to the cut-off values specified by the manufacturer.

## Data selection and exclusion criteria

This multicenter study utilized routine laboratory data from adult individuals (aged 18–50 years) who underwent thyroid function testing, including serum TSH, $fT_3$, $fT_4$.

   The dataset was initially screened to exclude individuals based on the following criteria:

(i)   missing or incomplete hormone data,

(ii)  documented diagnosis of cirrhosis, pulmonary failure, or chronic kidney disease,

(iii) confirmed pregnancy status in female participants,

(iv) patients with a documented diagnosis of thyroid disease based on hospital LIS–HIS records and/or those using medications known to affect thyroid function (e.g., levothyroxine, methimazole, propylthiouracil),

(v)  inpatients, including those from intensive care units, endocrinology, nephrology, and emergency departments (**Fig 1**).

   Subgroup stratification was performed according to gender (male/female) and thyroid antibody status. For antibody-negative status, individuals with both antibodies below assay cutoffs were included. For antibody-positive groups, at least one antibody needed to exceed the defined threshold.

   Outliers were detected separately within each subgroup using the Tukey method and the proportion of removed observations did not exceed 5% per subgroup. Each analyzer platform (e.g., Siemens, Abbott, Roche, Beckman) was processed and analyzed independently.

## Ethical principles

This study was conducted retrospectively using medical records and laboratory data collected from participating hospitals and laboratories. All data were fully anonymized before analysis. This study was approved by the Acıbadem Mehmet Ali Aydınlar University Medical Research Evaluation Board (ATADEK) with decision number 2023/07 dated April 28, 2023.

## Statistical analysis

After outlier exclusion, the distribution of each hormone within subgroups was assessed using the Shapiro–Wilk test. Since not all subgroups satisfied the assumption of normality (p < 0.05), group comparisons were consistently performed using the non-parametric Mann–Whitney U test for TSH, $fT_3$, and $fT_4$.

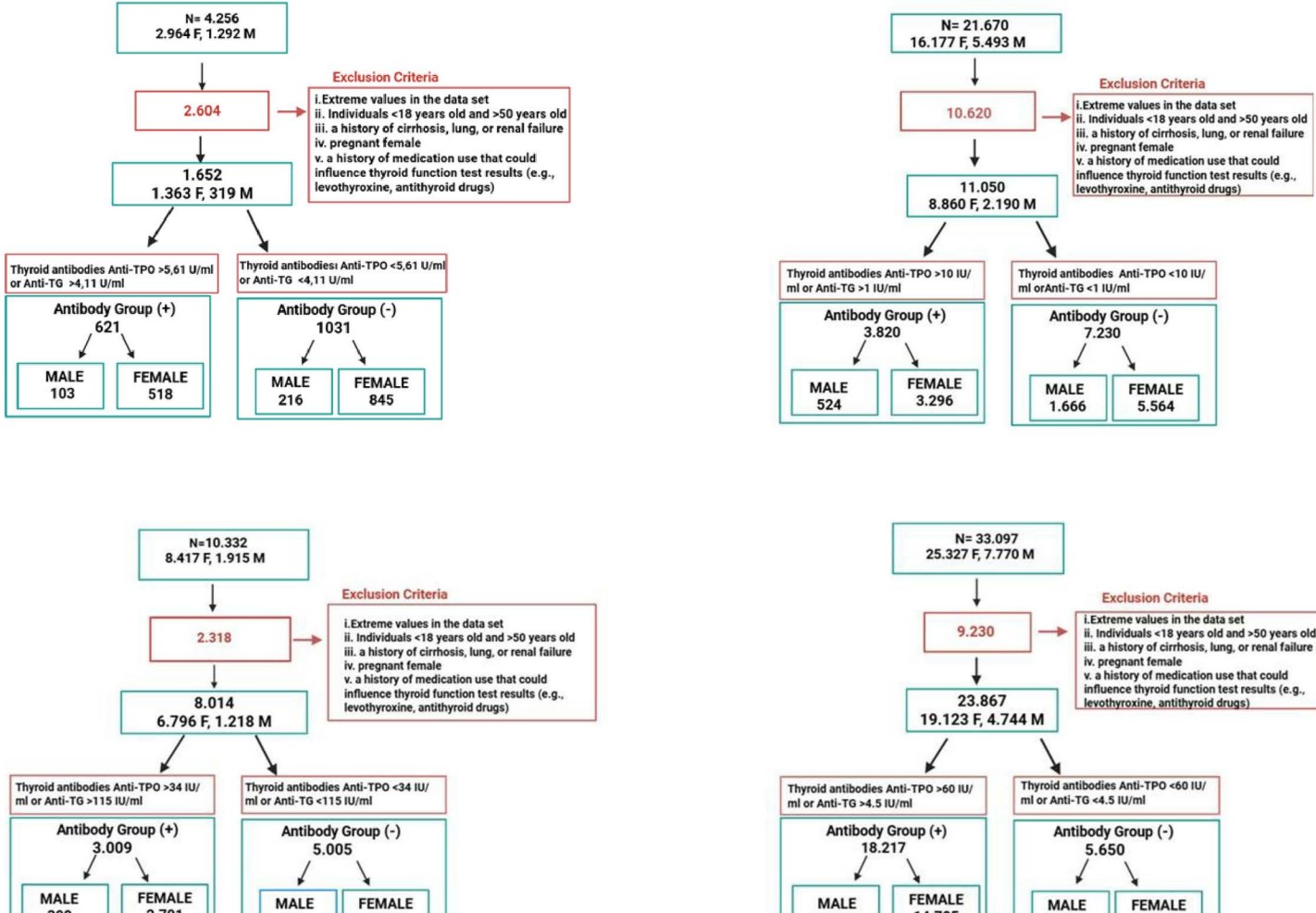

**Fig 1. Flow diagrams for participants examined using Abbott kits (upper left panel), Beckman kits (upper right panel), Roche kits (lower left panel), and Siemens kits (lower right panel).**

Reference intervals (RIs) for thyroid hormones were estimated using the refineR algorithm. RefineR is provided as an open-source R package (https://CRAN.R-project.org/package=refineR) and has been described and evaluated in detail in previous studies [21]. The method applies a Box–Cox transformation to approximate normality, followed by iterative exclusion of outliers and density-based modeling to isolate the central distribution. The final reference limits are obtained by determining the central 95% interval of this modeled distribution and back-transforming the results to the original scale.

The refineR algorithm was applied separately to each defined subgroup using 250 bootstrap replicates. For each subgroup, the lower and upper limits of the reference interval were determined along with the 90% confidence intervals of those limits. A 90% confidence interval was selected to maintain narrower interval widths, improving the detection of subgroup differences, in line with bootstrap-based RI methodology. The choice of 250 replicates was based on the original refineR publication, which used this number as a practical balance between computational cost and statistical robustness [21].

To compare the reference intervals between subgroups (e.g., female Ab(−) vs. female Ab(+)), the bias ratio (BR) was calculated for both lower and upper limits. The bias ratio is a standardized measure used to quantify the difference

between two reference interval limits by expressing the difference relative to the width of a reference interval. It is particularly useful for evaluating whether observed differences between subgroups are statistically and clinically meaningful. The bias ratio was calculated using the following formula:

$$BR_{\{LL\}} = \frac{(LL - LL^0)}{SD_{\{RI\}}} \ , \ BR_{\{UL\}} = \frac{(UL - UL^0)}{SD_{\{RI\}}}, \ SD_{\{RI\}} = \frac{UL^0 - LL^0}{3.92}$$

Where LL and UL represent the lower and upper limits of the subgroup being tested, and $LL_0$ and $UL_0$ refer to the corresponding limits in the reference group. The denominator, $SD_{\{RI\}}$ corresponds to the standard deviation across the reference interval. A |BR| value ≥ 0.375 was considered indicative of a statistically significant difference, as recommended by the IFCC C-RIDL and applied in previous validation studies [22]. All statistical analyses were performed using R version 4.4.1 (R Foundation for Statistical Computing, Vienna, Austria).

## Results

The lower and upper limit RI and (CI 95%) determined according to thyroid antibody levels in the four autoanalyzers are summarized in Tables 1–3 for serum TSH, $fT_4$, and $fT_3$, respectively.

In addition, Figs 1–3 show comparison graphs of the thyroid hormone RI and CI values calculated separately for individuals categorized according to thyroid Ab(+) and Ab(-) group for serum TSH, $fT_4$, and $fT_3$, respectively.

The statistical differences in the LL and UL RI limts of thyroid hormones, stratified by antibody status, were evaluated using the BR method.

### Serum TSH reference intervals according to age, gender, and antibody groups

$TSH_{(Abbott)}$ ın males, the RIs determined by antibody status (M− and M+) showed statistically significant differences in both the lower limit (LL: S) and upper limit (UL: S). In females, no significant difference was observed in the LL (LL: NS), but the UL differed significantly (UL: S) between antibody-negative (F−) and antibody-positive (F+) groups.

$TSH_{(Beckman)}$ ın males, the RI comparison based on antibody levels (M− vs. M+) revealed no significant difference in the LL (LL: NS), while the UL showed a considerable difference (UL: S). Similarly, in females, the LL did not differ significantly (LL: NS), but the UL exhibited a significant variation (UL: S) between F− and F+ groups.

$TSH_{(Roche)}$ ın males, significant differences were observed in both the LL and UL (LL: S, UL: S) when comparing antibody-negative (M−) and antibody-positive (M+) subgroups. In females, while the LL showed no significant difference (LL: NS), the UL was significantly different (UL: S) between F− and F+ groups.

$TSH_{(Siemens)}$ ın males, the RI analysis demonstrated no significant difference in the LL (LL: NS) but a significant difference in the UL (UL: S) between M− and M+ groups. In females, the LL was not significantly different (LL: NS), whereas the UL showed a significant variation (UL: S) between F− and F+ individuals (**Table 1**, **Fig 2**).

### Serum $fT_4$ reference intervals according to age, gender, and antibody groups

$fT_{4(Abbott)}$ ın both gender (M − vs. M+ and F − vs. F+), the RIs showed no significant difference in the LL (LL: NS) but a significant difference in the UL (UL: S).

$fT_{4(Beckman)}$ ın both gender (M − vs. M+ and F − vs. F+), the RIs showed no significant difference in the LL (LL: NS) but a significant difference in the UL (UL: S).

$fT_{4(Roche)}$ ın males, significant differences were observed in both LL and UL (LL: S, UL: S). In females, the LL differed significantly (LL: S), but the UL showed no significant difference (UL: NS).

$fT_{4(Siemens)}$ no significant differences were detected in either the LL or UL (LL: NS, UL: NS) for any subgroup (M − /M + , F − /F+) (**Table 2**, **Fig 3**).

**Table 1. Reference and CI 95% determined for TSH depending on and independent of antibody levels.**

| Analyzer | Gender | Antibody Group | N | TSH mIU/l | 90% Confidence İnterval | Bias Ratio |
|---|---|---|---|---|---|---|
| Architect i2000SR Manufacturer RI | M | – | 216 | 0.25–4.08 | 0.12-0.42- 2.02-5.95 | LL*+UL* |
| | M | + | 103 | 1.20–3.08 | 0.86-1.49 - 2.79-4.61 | |
| | F | – | 845 | 0.24–3.50 | 0.16-0.44 - 2.64-6.01 | LL+UL* |
| | F | + | 518 | 0.41–7.44 | | 0.15-0.53 - 3.05-7.55 |
| | M | +,- | 320 | 0.16–5.18 | 0.11-1.05 - 2.88-6.30 | LL+UL* |
| | F | +,- | 1364 | 0.32–6.15 | | 0.18-0.50 - 2.98-6.82 |
| | M+F | – | 1061 | 0.34–6.38 | 0.18-0.44 - 2.86-6.80 | LL+UL* |
| | M+F | + | 621 | 0.39–7.01 | | 0.15-0.59 −3.81-7.50 |
| | M+F | +,- | 1683 | 0.36–6.56 | 0.19-0.61 - 4.18-7.32 | LL+UL* |
| | | | | | 0.35-4.94 | |
| DxI 800 Unicel Manufacturer RI | M | – | 1666 | 0.39–3.96 | 0.27-0.63- 2.93-4.90 | LL+UL* |
| | M | + | 524 | 0.12–5.23 | 0.11-0.47- 4.49-7.31 | |
| | F | – | 5564 | 0.41–4.77 | 0.3-0.48- 4.20- 4.99 | LL+UL* |
| | F | + | 3296 | 0.45–6.48 | | 0.18-0.67 - 4.28-8.01 |
| | M | +,- | 2166 | 0.30–4.17 | 0.24-0.63 - 3.87-5.74 | LL+UL* |
| | F | +,- | 8838 | 0.44–5.31 | | 0.26-0.52 - 4.49-5.66 |
| | M+F | – | 7229 | 0.47–4.78 | 0.34-0.53- 4.14-5.01 | LL+UL* |
| | M+F | + | 3812 | 0.40–6.3 | | 0.17-0.65- 4.34-7.96 |
| | M+F | +,- | 11010 | 0.46–5.33 | 0.31-0.54 - 4.38-5.64 | LL+UL |
| | | | | | 0.38-5.33 | |
| Modular E170 Manufacturer RI | M | – | 910 | 0.44–3.63 | 0.29-0.68 - 2.53-4.33 | LL*+UL* |
| | M | + | 308 | 0.19–5.77 | 0.11-0.71 - 3.38-7.65 | |
| | F | – | 4095 | 0.42–4.68 | 0.28-0.47 - 2.93-4.79 | LL+UL* |
| | F | + | 2701 | 0.65–7.31 | | 0.22-0.68 - 3.76-7.42 |
| | M | +,- | 1201 | 0.45–3.98 | 0.23-0.63 - 2.72-5.17 | LL+UL* |
| | F | +,- | 6755 | 0.45–5.25 | | 0.33-0.54 - 4.35-5.82 |
| | M+F | – | 5007 | 0.44–4.60 | 0.3-0.53 - 2.79-4.93 | LL+UL* |
| | M+F | + | 3010 | 0.66–8.14 | | 0.2-0.67 - 3.72-8.14 |
| | M+F | +,- | 7961 | 0.60–6.08 | 0.38-0.63 - 4.48-6.22 | LL+UL* |
| | | | | | 0.27-4.20 | |
| Atellica IM Manufacturer RI | M | – | 1232 | 0.52–4.36 | 0.30-0.54- 2.56-4.44 | LL+UL* |
| | M | + | 3512 | 0.52-4.51 | 0.27-0.58 - 2.81-5.15 | |
| | F | – | 4418 | 0.51–3.37 | 0.41-0.58 - 2.81-4.50 | LL+UL* |
| | F | + | 14705 | 0.59–5.58 | | 0.43-0.59 - 4.17-5.60 |
| | M | +,- | 4739 | 0.54-4.81 | 0.29-0.56 - 2.65-4.98 | LL+UL |
| | F | +,- | 19125 | 0.58–5.25 | | 0.52-0.61 - 4.72-5.44 |
| | M+F | – | 5653 | 0.57–4.53 | 0.42-0.59 - 2.94-4.71 | LL+UL* |
| | M+F | + | 18233 | 0.56–5.29 | | 0.49-0.59 - 4.76-5.64 |
| | M+F | +,- | 23866 | 0.58–5.3 | 0.54-0.60 - 4.93-5.37 | LL+UL |
| | | | | | 0.55-4.78 | |

TSH;thyrotropin, + ,-; Independent of antibody, M + F; Independent of gender, LL: lower limit, UL;upper limit, * Significant Difference [25]

**Table 2. Reference and CI 95% determined for fT$_4$ depending on and independent of antibody levels.**

| Analyzer | Gender | Antibody Group | N | fT$_4$ (pmol/l) | 90% Confidence İnterval | Bias Ratio |
|---|---|---|---|---|---|---|
| Architect i2000SR Manufacturer RI | M | – | 220 | 9.98–14.64 | 9.59-10.91 - 13.4-15.13 | LL+UL |
| | M | + | 102 | 9.70-14.82 | 9.49-10.66 - 13.13-15.1 | |
| | F | – | 855 | 9.58–15.07 | 9.43-10.31 - 14.28-15.28 | LL+UL |
| | F | + | 515 | 9.14–14.86 | 8.97-10.76 - 13.24-15.3 | |
| | M | +,- | 322 | 9.46–15.12 | 9.41-10.55 - 13.47-15.26 | LL+UL |
| | F | +,- | 1368 | 9.45–15.18 | 9.29-9.94 - 14.45-15.36 | |
| | M+F | – | 1072 | 9.53–14.99 | 9.41-10.26 −14.31-15.2 | LL+UL |
| | M+F | + | 617 | 9.09–14.85 | 8.99-10.56 −13.43-15.37 | |
| | M+F | +,- | 1691 | 9.42–15.14 | 9.28-9.94 - 14.63-15.38 | LL+UL* |
| | | | | | 9.01- 19.02 | |
| DxI 800 Unicel Manufacturer RI | M | – | 1770 | 8.43–15.8 | 8.22-8.6 - 15.44-16.26 | LL*+UL |
| | M | + | 560 | 6.76–16.3 | 6.40-8.21 - 15.0-17.29 | |
| | F | – | 5891 | 8.31–15.65 | 8.11-8.39 - 15.24-15.85 | LL+UL |
| | F | + | 3522 | 7.7–16.08 | 7.56-7.91 - 15.75-16.56 | |
| | M | +,- | 2329 | 8.38–16.10 | 8.01-8.44 - 15.62-16.43 | LL+UL |
| | F | +,- | 9413 | 8.16–15.78 | 7.91-8.24 - 15.31-15.99 | |
| | M+F | – | 7661 | 8.33–15.73 | 8.18-8.4 - 15.34-15.91 | LL*+UL |
| | M+F | + | 4082 | 7.57–16.06 | 7.44-7.86 - 15.76-16.67 | |
| | M+F | +,- | 11741 | 8.08–15.61 | 7.95-8.23 - 15.35-15.98 | LL+UL* |
| | | | | | 7.86-14.42 | |
| Modular E170 Manufacturer RI | M | – | 921 | 12.89-19.99 | 12.61-13.89 −18.52-21.18 | LL*+UL* |
| | M | + | 305 | 12.10-23.03 | 11.6-14.99 - 17.75-22.72 | |
| | F | – | 4122 | 10.34-19.35 | 10.23-11.14 - 19.12-19.63 | LL*+UL* |
| | F | + | 2715 | 11.42–20.42 | 10.85-11.57 −19.29-20.7 | |
| | M | +,- | 1226 | 12.34-20.43 | 12.26-13.81 −18.99-21.73 | LL*+UL |
| | F | +,- | 6835 | 10.78-19.67 | 10.47-11.34 −19.28-20.11 | |
| | M+F | – | 5048 | 10.47-19.82 | 10.44-11.33 −19.55-20.19 | LL*+UL* |
| | M+F | + | 3021 | 11.46–20.81 | 10.91-11.62 −19.38-20.99 | |
| | M+F | +,- | 8069 | 11.23–20.35 | 10.67-11.45 −19.56-20.54 | LL+UL* |
| | | | | | 12.00-22.00 | |
| Atellica IM Manufacturer RI | M | – | 1258 | 12.30-21.05 | 12.01-13.23 −19.20-20.99 | LL+UL |
| | M | + | 3623 | 11.90–20.34 | 11.71-12.13 - 19.44-20.57 | |
| | F | – | 4487 | 11.43–18.64 | 11.3-11.73 - 18.04-18.78 | LL+UL |
| | F | + | 14949 | 11.20–18.78 | 11.06-11.25 - 18.07-18.81 | |
| | M | +,- | 4881 | 12.13-20.55 | 11.78-12.19 - 19.61-20.8 | LL*+UL* |
| | F | +,- | 19440 | 11.20–18.49 | 11.11-11.32 - 18.19-18.93 | |
| | M+F | – | 5751 | 11.43–18.95 | 11.34-11.76 −18.62-19.33 | LL+UL |
| | M+F | + | 18563 | 11.26–19.12 | 11.17-11.32 - 18.6-19.3 | |
| | M+F | +,- | 24314 | 11.28–18.93 | 11.21-11.37 −18.52-19.37 | LL+UL* |
| | | | | | 11.50-22.70 | |

fT$_4$; free thyroxine, +,-; Independent of antibody, M+F; Independent of gender, LL: lower limit, UL; upper limit, * Significant Difference [25]

**Table 3. Reference and CI 95% determined for $fT_3$ depending on and independent of antibody levels.**

| | Gender | Antibody Group | N | $fT_3$ (pmol/l) | 90% Confidence İnterval | Bias Ratio |
|---|---|---|---|---|---|---|
| Architect i2000SR Manufacturer RI | M | – | 218 | 2.41–3.47 | 2.27-2.8- 3.21-3.96 | LL*+UL |
| | M | + | 101 | 2.77–3.45 | 2.71-2.86 - 3.38-3.51 | |
| | F | – | 856 | 2.24–3.78 | 2.16-2.36 - 3.51-3.84 | LL+UL* |
| | F | + | 510 | 2.37–3.36 | 2.24-2.59 - 3.22-3.51 | |
| | M | +,- | 318 | 2.35–3.81 | 2.19-2.72 - 3.47-3.94 | LL+UL* |
| | F | +,- | 1369 | 2.25–3.67 | 2.12-2.34 - 3.46-3.7 | |
| | M+F | – | 1075 | 2.21–3.81 | 2.14-2.3 - 3.60-3.94 | LL+UL* |
| | M+F | + | 612 | 2.22–3.51 | 2.18-2.46 - 3.34-3.59 | |
| | M+F | +,- | 1689 | 2.22–3.7 | 2.09-2.32 - 3.55-3.8 | LL*+UL* |
| | | | | | 0.35-4.94 | |
| DxI 800 Unicel Manufacturer RI | M | – | 1771 | 2.82–4.65 | 2.74-2.87- 4.54-4.87 | LL+UL* |
| | M | + | 556 | 2.65–4.83 | 2.58-2.82- 4.60-5.18 | |
| | F | – | 5879 | 2.60–4.35 | 2.56-2.71- 4.05-4.45 | LL+UL |
| | F | + | 3504 | 2.51–4.43 | 2.49-2.57- 4.23-4.52 | |
| | M | +,- | 2326 | 2.79–4.73 | 2.74-2.87- 4.57-4.88 | LL*+UL* |
| | F | +,- | 9382 | 2.58–4.26 | 2.53-2.6 - 4.10-4.46 | |
| | M+F | – | 7647 | 2.62–4.28 | 2.58-2.67- 4.20-4.56 | LL*+UL* |
| | M+F | + | 4066 | 2.52–4.56 | 2.5-2.58- 4.35-4.62 | |
| | M+F | +,- | 11711 | 2.61–4.46 | 2.56-2.62- 4.34-4.57 | LL*+UL* |
| | | | | | 0.38-5.33 | |
| Modular E170 Manufacturer RI | M | – | 921 | 4.08–6.45 | 4.02-4.53- 6.19-6.54 | LL+UL |
| | M | + | 299 | 4.12–6.45 | 4.05-4.8-5.49-6.43 | |
| | F | – | 4137 | 3.54–5.93 | 3.48-3.69- 5.73-5.97 | LL+UL |
| | F | + | 2726 | 3.58–5.78 | 3.45-3.64- 5.51-5.81 | |
| | M | +,- | 1221 | 4.05–6.37 | 4.03- 4.51 - 6.14-6.56 | LL*+UL* |
| | F | +,- | 6863 | 3.53–5.84 | 3.48-3.57- 5.64-5.95 | |
| | M+F | – | 5065 | 3.56–6.16 | 3.52-3.69- 5.95-6.2 | LL+UL* |
| | M+F | + | 3027 | 3.48–5.66 | 3.46-3.65 - 5.56-5.96 | |
| | M+F | +,- | 8095 | 3.57–6.13 | 3.52-3.58- 5.85-6.15 | LL*+UL* |
| | | | | | 0.27-4.20 | |
| Atellica IM Manufacturer RI | M | – | 1253 | 4.29–6.73 | 4.20-4.54 - 6.46-6.82 | LL+UL* |
| | M | + | 3607 | 4.28–6.30 | 4.22-4.44 - 6.23-6.41 | |
| | F | – | 4500 | 3.86–6.09 | 3.74-3.98 - 5.93-6.13 | LL+UL* |
| | F | + | 14991 | 3.77–5.76 | 3.73-3.86 - 5.70-5.82 | |
| | M | +,- | 4861 | 4.22–6.36 | 4.2-4.40 - 6.30-6.5 | LL*+UL* |
| | F | +,- | 19486 | 3.79–5.84 | 3.73-3.86 −5.76-5.91 | |
| | M+F | – | 5752 | 3.94–6.33 | 3.77-3.99 −6.14-6.36 | LL+UL* |
| | M+F | + | 18620 | 3.84–6.03 | 3.77-3.86- 5.93-6.06 | |
| | M+F | +,- | 24368 | 3.86–6.13 | 3.77-3.88 - 5.98-6.14 | LL*+UL* |
| | | | | | 0.55-4.78 | |

$fT_3$; free triiodothyronine, +,-; Independent of antibody, M+F; Independent of gender, LL: lower limit, UL; upper limit, * Significant Difference [25]

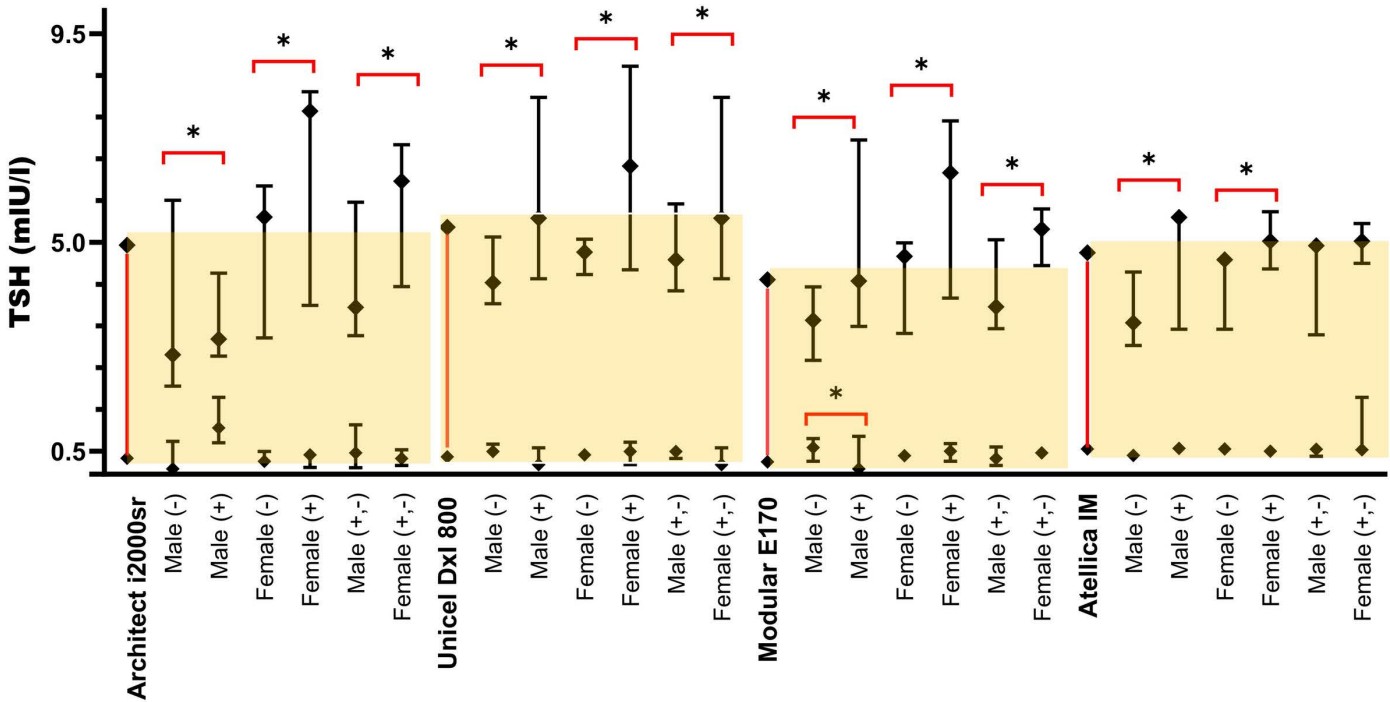

**Fig 2. Comparative Reference and CI 95% for TSH in Different Autoanalyzers.** *The yellow background indicates the reference intervals recommended by each manufacturer.

### Serum fT₃ reference intervals according to age, gender, and antibody groups

fT$_{3(Abbott)}$ in males, significant differences were observed in the lower limit (LL: S) of the RI between antibody-negative (M-) and antibody-positive (M+) groups, while the LL (UL) showed no significant difference (UL: NS). In females, both LL and UL demonstrated statistically significant differences (LL: S, UL: S) between F- and F+ groups.

fT$_{3\ (Beckman)}$ in males, significant differences were observed in the lower limit (LL: S) of the RI between antibody-negative (M-) and antibody-positive (M+) groups, while the upper limit (UL) showed no significant difference (UL: NS). Among female subjects, both LL and UL demonstrated statistically significant differences (LL: S, UL: S) between F- and F+ groups.

fT$_{3\ (Roche)}$ in males, showed no significant differences in either RI limit (LL: NS, UL: NS) when comparing antibody status groups. Similarly in females, significant variation was observed only in the LL (LL: S), with no significant difference in the UL (UL: NS).

fT$_{3\ (Siemens)}$ in males, significant differences were detected in the LL (LL: S) but not in the UL (UL: NS) of the RI. In females, demonstrated no significant differences in either RI limit (LL: NS, UL: NS) between antibody status groups (**Table 3**, **Fig 4**)

### Serum TSH reference intervals independent of antibody levels

The TSH, fT₄, and fT₃ values calculated independently of thyroid antibody levels in females and males are presented in Tables 1, 2, and 3, respectively. Additionally, comparison graphs showing the RI and CI values for thyroid hormones, calculated independently of thyroid antibody levels, are presented for TSH, fT₄, and fT₃ in Figs 2, 3, and 4, respectively.

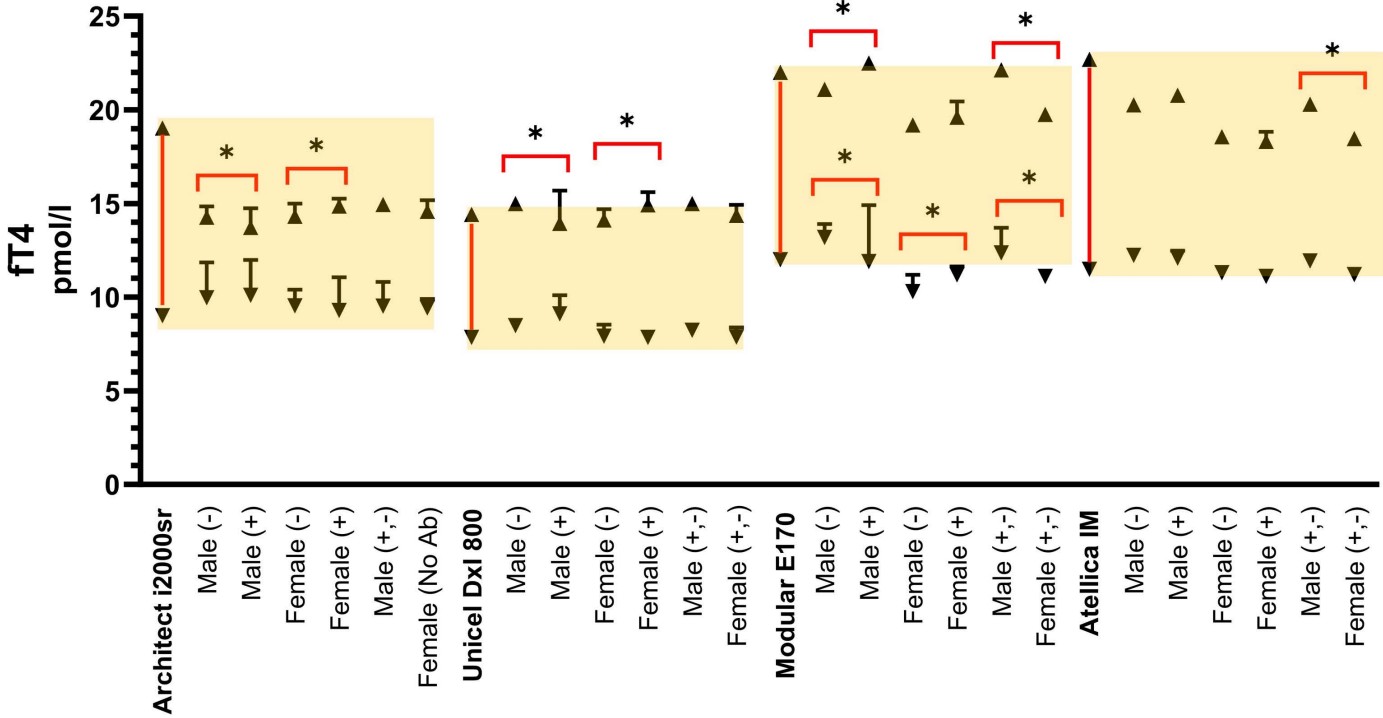

**Fig 3. Comparative Reference İntervals and CI 95% for fT$_4$ in Different Autoanalyzers.** *The yellow background indicates the reference intervals recommended by each manufacturer.

The statistical differences between LL and UL RI limits for thyroid hormones, determined independently of antibody status, were evaluated using the BR method.

### Serum TSH reference intervals independent of antibody levels

When comparing gender-specific RI independent of thyroid antibody status (M+,- F+,-), significant differences were observed in the UL(UL: S) but not in the LL (LL: NS) for TSH$_{(Abbot, Beckman, Roche)}$. However, TSH$_{(Siemens)}$ showed with no significant gender-based differences in either limit (**Table 1**, **Fig 2**).

### *Serum fT$_4$ reference intervals independent of antibody levels*

fT$_{4(Roche)}$, when comparing gender-specific RI independent of thyroid antibody status (M+,- F+,-), statistically significant differences were observed in both the LL and UL (LL: S, UL: S).

In contrast, fT$_{4(Abbott, Beckman)}$ revealed no significant gender-based differences in either the LL or UL RI (LL: NS, UL: NS). fT$_{4(Siemens)}$with no significant gender difference in the LL (LL: NS) but a statistically significant difference in the UL (UL: S) (**Table 2**, **Fig 3**).

### *Serum fT$_3$ reference intervals independent of antibody levels*

fT$_{3(Beckman, Roche, Siemens)}$ istatistically significant gender differences were observed in both the LL (LL: S) and UL (UL: S) of RIs when comparing males and females independent of thyroid antibody status (M+,- F+,-). In contrast, fT$_{3(Abbott)}$ showed no significant gender difference in the LL (LL: NS) but demonstrated a statistically significant difference in the UL (UL: S) (**Table 3**, **Fig 4**).

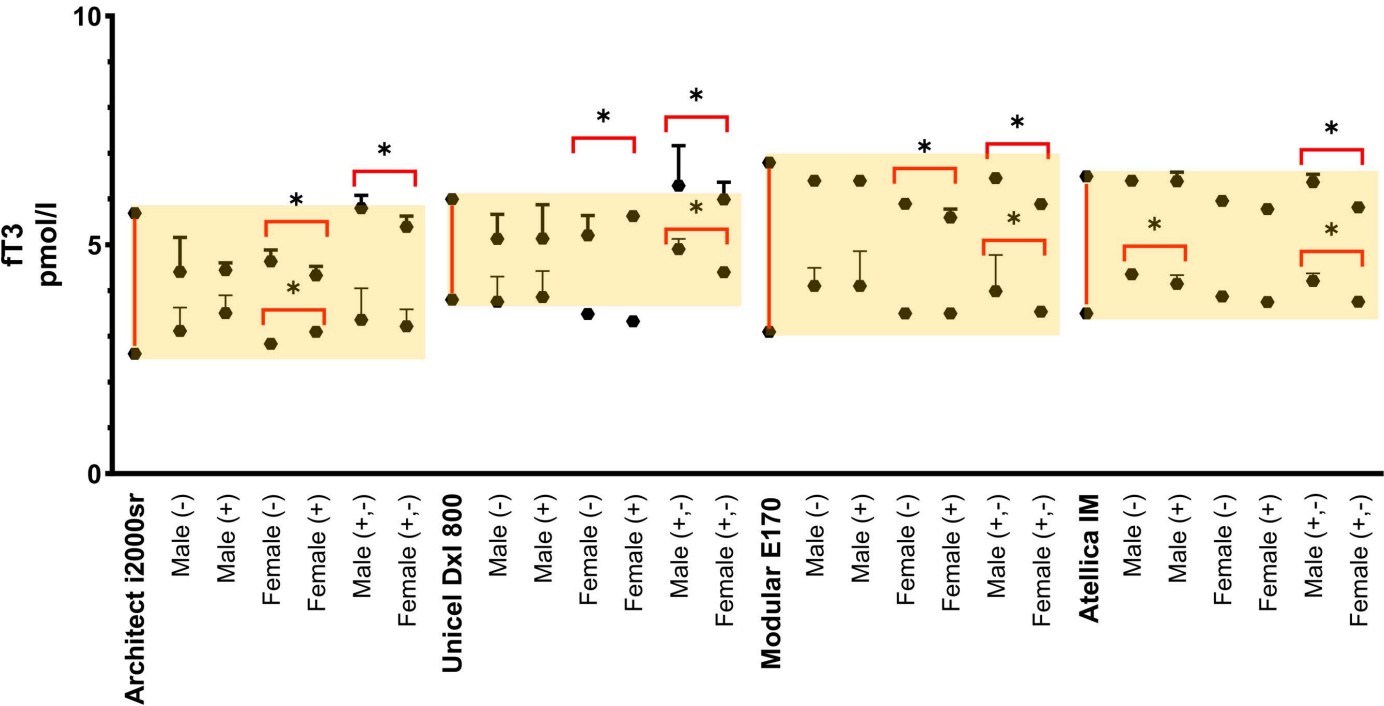

**Fig 4. Comparative Reference Intervals and CI 95% for fT$_3$ in Different Autoanalyzers.** *The yellow background indicates the reference intervals recommended by each manufacturer.

## Discussion

In this study, the influence of variables such as thyroid antibody status and gender on the RI of TSH, fT$_4$ and fT$_3$ hormones was systematically investigated in the Turkish population aged 18–50 years using four different autoanalyzers. Our findings demonstrate that, particularly for TSH, RI vary significantly depending on both thyroid antibody levels and gender. These results highlight the necessity of considering both antibody status and demographic variables when establishing RI for thyroid function tests, and clearly underscore the need for standardization across different analytical platforms.

### Evaluation of thyroid hormone reference intervals based on thyroid antibody status and gender

In the current literature, most studies defining RI for thyroid hormones typically present uniform RI for the general population, often disregarding thyroid antibody status. However, previous studies have emphasized that elevated anti-TPO and anti-Tg levels are significant risk factors for the development of subclinical hypothyroidism [23]. Despite this, the impact of antibody status on RIs has not been adequately investigated to date. Our study addresses this critical gap in the literature by being one of the few multicenter studies to establish antibody-specific RI for thyroid hormones across four different autoanalyzers.

The accurate determination of RI plays a critical role in the clinical interpretation of thyroid function tests. Our most striking finding was the consistent and statistically significant difference in TSH RI across analyzers according to antibody status. For instance, our study demonstrated distinct RI for males when stratified by antibody levels on both TSH (Abbott, Roche) autoanalyzers. Furthermore, TSH RI determined by thyroid antibody levels is higher in females than in males (e.g., Roche analyzer: Ab- females 0.42–4.68 mU/L vs. males 0.44–3.63 mU/L). This finding further highlights the role of gender hormones in the regulation of TSH.

 

One of the major strengths of our study is its multicenter design, conducted across different regions of Turkey. The inclusion of metropolitan cities such as Ankara and Istanbul, which reflect the general demographic structure of the country, enhances the representativeness of our findings for the broader Turkish population. Notably, in regions with endemic goiter, TSH RI were significantly affected in both antibody-negative and antibody-positive individuals. This finding suggests that the standard RI recommended by current guidelines (e.g., 4.0–5.0 mU/L) may be inadequate for Ab+ populations.

Another important finding of our study is the observed variation in RI among different autoanalyzers, despite the use of a standardized algorithm (RefineR) for RI estimation across all platforms. The discrepancies between devices underscore the necessity of analytical harmonization. For instance, the Beckman and Abbott analyzers demonstrated greater sensitivity to antibody status in $fT_3$ measurements, while the Roche analyzer showed the most prominent RI variation for $fT_4$. Additionally, the Roche analyzer yielded the widest TSH interval for Ab+ males, whereas the Abbott analyzer provided the narrowest. These differences are likely due to variations in measurement methodologies, calibration protocols, and reagent compositions. From a methodological perspective, the consistent observation of this effect across all four analyzers (Abbott, Beckman, Roche, Siemens) supports the reliability of our findings. However, the observed inter-analyzer variability further emphasizes the critical need for laboratories to establish their own population-specific RI rather than relying solely on manufacturer-provided or generalized RI.

Our findings contribute to clinical practice from three distinct perspectives:

(i)   Antibody status should always be taken into account when interpreting thyroid function tests.

(ii)  Particularly in Ab+ female patients, the currently used reference intervals (RI) may be inadequate.

(iii) In countries like Turkey, where endemic goiter is prevalent, laboratories should establish region- and analyzer-specific RI.

International organizations such as ATA and the IFCC have also recommended the inclusion of gender as a factor in the interpretation of thyroid function tests [12,17,18]. Our findings support these recommendations and further emphasize the importance of considering both antibody status and gender in clinical interpretation. In particular, broader TSH intervals observed among Ab+ individuals suggest that autoimmune processes may affect gender differently. Therefore, the use of a uniform RI may result in misdiagnoses and unnecessary interventions.

In conclusion, this study highlights the necessity of considering analyzer-specific differences, thyroid antibody status, and gender when establishing RI for thyroid function tests. Even when using the same algorithm (RefineR), RI values varied significantly across different analyzers, indicating that such differences may have clinical relevance beyond mere analytical variability. This finding underscores the importance of inter-analyzer variation not only from an analytical perspective but also in clinical interpretation.

The need for harmonization and standardization across analytical platforms has long been emphasized in the literature [24–29]. Our study supports the notion that this need must also be addressed during RI determination processes. Furthermore, the consistently elevated TSH levels observed in Ab+ individuals indicate that incorporating antibody status into RI determination may enhance diagnostic accuracy. Ignoring these differences, especially in patients with borderline hormone levels, could potentially lead to misdiagnosis or inappropriate treatment decisions.

Overall, our findings suggest that developing personalized RI accounting for analyzer-specific characteristics as well as individual variables such as gender and antibody status may significantly improve the diagnostic reliability of thyroid function testing.

### Evaluation of thyroid hormone reference ıntervals: comparison with manufacturer data and literature ındependent of thyroid antibody status

In the second phase of this study, RI were established separately for males and females without considering thyroid antibody status. In clinical practice, physicians often rely on manufacturer-recommended RI, which serve as the basis for

diagnosis and treatment decisions. However, it has long been debated to what extent these reference values reflect the characteristics of local populations [30]. In this section of the discussion, the RI established without taking antibody status into account were evaluated in comparison with both population-based studies in the literature and the reference values provided by manufacturers, which are often based on American or European populations.

The findings demonstrated that gender significantly influences thyroid hormone levels. Notably, higher upper limits of TSH were observed in females, underscoring the importance of using gender-specific RIs in the diagnosis of subclinical hypothyroidism. In alignment with literature reporting greater variability in thyroid hormone levels among females [31,32] this study also showed clear effects of gender on reference intervals. Similarly, the variability of $fT_4$ and $fT_3$ levels across different analyzers highlights the necessity of reporting gender-specific and locally derived reference values in laboratory reports. These findings support the recommendations of international organizations such as ATA and IFCC regarding the use of population-specific RI in thyroid testing [12,17,18].

Gender-based differences also varied according to the autoanalyzer used. Our results revealed significant gender-specific differences in TSH RI across all analyzers. For TSH $_{(Abbott, Beckman, Siemens, Roche)}$ wider RI were found in females compared to males. For $fT_{4(Abbott)}$, no significant gender difference was observed, while in Roche and Beckman analyzers, males exhibited broader RI. For $fT_{3(Abbott, Beckman, Roche, Siemens)}$ males generally showed wider intervals. These results emphasize the importance of considering both analyzer-specific and gender-related differences to improve diagnostic accuracy in thyroid function testing.

Another important finding was the inconsistency between RI calculated without considering antibody status and those provided by manufacturers based on American/European populations. This discrepancy may be attributed to factors such as iodine intake, genetic variability, and ethnic differences. For example, the UL of TSH calculated for females using the Abbott analyzer (7.44 mIU/L) was considerably higher than the manufacturer's recommended value (4.94 mIU/L). This difference suggests that individuals who might be diagnosed with hypothyroidism in clinical practice could in fact be within physiological limits, raising concerns about overdiagnosis.

In conclusion, our study demonstrates that manufacturer-provided RI often derived from American and European populations may fail to adequately reflect local biological variability. Furthermore, our findings are consistent with previous literature highlighting population-specific and gender-related differences in thyroid hormone RI [12,20,21,33].These results suggest that laboratories should consider developing their own locally derived reference intervals rather than relying solely on manufacturer data, thereby ensuring more accurate and reliable clinical decision-making.

In this section of the discussion, the RI determined independently of antibody levels were compared with existing population-based studies in the literature.

### Architect i2000sr (Abbott Laboratories, Abbott Park, Illinois, ABD)

In studies conducted by Motor et al. (2010) and Örkmez et al. (2023), narrower TSH RIs were reported for females using the Architect i2000sr (Abbott Laboratories, Abbott Park, Illinois, USA) analyzer [34,35]. However, in our study, we identified wider TSH RI for females. These discrepancies may stem from differences in sample size, age distribution, and methodological approaches. Although the manufacturer's recommended ranges are generally broader, we observed narrower RI for $fT_4$ and $fT_3$ in our study, and gender-specific differences for TSH were also evident when using the Abbott autoanalyzer.

Studies using the same autoanalyzer in different populations have reported significant variations in the lower and upper limits of thyroid hormone RI [33,36,37]. Therefore, the discrepancies observed between our findings and those in the literature highlight the necessity for each country to establish its own population-specific RI.

### Modular E170 Analizör (Roche Diagnostics, Almanya)

Gender-specific differences in TSH RI observed in our study contradict the findings reported by Yıldız et al. (2022) [38]. However, another study emphasized the diagnostic relevance of gender-specific RI, thus supporting our results [12]. Regional studies conducted in China and Korea validated our established $fT_4$ RI for both males and females. However,

these studies reported wider TSH RI compared to our findings [14,39]. These results illustrate the critical influence of demographic and methodological factors in determining RI. Furthermore, the RI established in our study differed from those recommended by the manufacturer.

Lewis et al. reported no clinically meaningful differences in TSH measurements between Roche Cobas and Siemens Atellica platforms and therefore supported the use of harmonized TSH reference intervals [40]. In contrast, a substantial positive inter-analyzer bias was observed for fT4, precluding the recommendation of harmonized reference intervals for this analyte.

Our findings are consistent with recent large-scale studies employing similar methodological approaches. In the study by Bohn et al., reference intervals for thyroid hormones were derived using the refineR method across multiple autoanalyzers and large population datasets, demonstrating methodological similarities to our approach [41]. However, an important distinction is that thyroid autoantibody status was not considered as a stratification variable during reference interval derivation in that study. Their findings demonstrated that due to significant inter-analyzer differences, harmonized RI were not recommended for free thyroxine (fT4), whereas no statistically significant differences were observed between analyzers for thyroid-stimulating hormone (TSH).

In contrast to these important studies, our findings reveal a more nuanced picture; RI for thyroid function tests are influenced not only by the analytical platform but also significantly by thyroid antibody status and demographic factors. These results support the necessity for establishing analyzer-specific and autoantibody-stratified RI that are validated at the local population level, thereby providing a more precise and clinically relevant approach to thyroid function test interpretation.

### Atellica IM (Siemens Diagnostics, Tarrytown, NY)

TSH and $fT_4$ RI identified in our study were narrower than those reported by Kösoğlu et al. (2010) but consistent with the ranges reported by Enli et al. (2004) [42,43]. In contrast, our $fT_3$ RI were broader than those reported in both studies. When compared to studies conducted in Polish and Serbian populations, discrepancies were observed, likely due to differences in indirect RI determination methods and demographic/ethnic variations [44,45]. These findings underscore the importance of using standardized methodologies and establishing population-specific RI.

### Access DxI 800 Unicel (Beckman Coulter, ABD)

The LL of the TSH RI in our study were consistent with those reported by Çelebiler et al. (2010), while the upper limits were found to be broader [33]. Similarly, a study conducted in Italy reported comparable upper limits for TSH in females. In contrast, a study from the Pakistani population reported broader $fT_4$ RI compared to our findings [46]. These discrepancies may be attributed to differences in sample size, age range, and demographic characteristics, all of which should be considered when interpreting results.

In our study, the RI for thyroid hormones determined using four different automated analyzers revealed significant discrepancies between autoanalyzer. These differences can primarily be attributed to the methodological characteristics of immunoassays, particularly the specificity and sensitivity of the antibodies used. The literature indicates that variations in assay methodologies, even among healthy populations, can influence the reference intervals and subsequently affect clinical interpretation [47,48]. For instance, in a study conducted by Barth et al. (2020) following the IFCC-CRIDL protocol, samples from healthy individuals without systemic disease were analyzed using different autoanalyzer, and significant methodological differences were identified between the autoanalyzer [49]. Similarly, in a more comprehensive study involving eight different immunoassay kits, although a high correlation was observed across kits (R>0.99), the slopes varied between 0.75 and 1.06. This finding highlights that thyroid hormones, especially TSH, can vary significantly depending on the kit used, which may affect medical decision thresholds [50]. These results underscore that, even in healthy individuals, the choice of autoanalyzer or assay kit can influence RI and thereby affect clinical decision-making. Furthermore, the IFCC emphasizes that different autoanalyzer and methodologies used in immunoassay testing can have a significant impact on

measurement results [51]. The IFCC working group on hormone assay harmonization highlights the global need for standardization to ensure comparability of hormone measurements such as TSH across laboratories and analytical platforms.

### Limitations

This study has several limitations that should be acknowledged. Firstly, while the 90% CI of the RI determined for different groups using the Atellica IM autoanalyzer showed notable proximity, increasing the number of bootstrapping iterations to 800 did not yield significant differences in the results.

Secondly, the age range of 18–50 years was selected as the primary focus for determining reference intervals due to its widespread use in clinical practice and the characteristics of our current dataset. However, we recognize the need to expand our research to include individuals both under 18 years and over 50 years of age. Ongoing studies are being conducted to address this limitation and provide a more comprehensive understanding of thyroid function across all age groups.

Thirdly, our study population included a higher number of female participants compared to males. This reflects the higher prevalence of thyroid disease among female, which is a well-documented epidemiological feature. While this gender imbalance is representative of the clinical reality, it may have influenced the generalizability of our findings, particularly in male subgroups. Future studies with a more balanced gender distribution are needed to validate and refine these reference intervals further.

Fourthly, as this study was conducted over several years, reagent lot-to-lot variation and seasonal variability may have influenced test results. While the same analyzers were used across all centers, the use of different reagent lots is considered a potential limitation of the study.

Fifthly, due to national regulations, reflex testing for the assessment of thyroid function could not be applied, and we consider this a limitation of the study.

Finally, iodine status was not directly measured in the patient population, and iodine measurements were not included as a variable in the analysis. Although iodine supplementation is provided through salt fortification and dietary sources in our country, iodine deficiency still exists in the population.

### Conclusion

In this study, we showed that thyroid antibody status and gender significantly influenced thyroid hormone RI across four different autoanalyzers. These findings highlight the importance of considering both thyroid antibody status and gender when determining RI for thyroid function tests, particularly for TSH levels, in clinical practice. To date, most RI studies have been conducted without measuring thyroid antibody levels, which may lead to inaccurate or misleading RI. Therefore, we recommend that future prospective studies focusing specifically on TSH measure thyroid antibody levels and determine RI based on these measurements.

Additionally, our study revealed significant variability in thyroid hormone RI across the four autoanalyzers, even when using the same algorithm (RefineR). These differences emphasize the challenges in standardizing thyroid function tests and suggest that such variability may have important implications for clinical practice and patient management. Clinicians should be aware of potential discrepancies between autoanalyzers when interpreting thyroid function test results, as these differences could influence diagnostic accuracy and treatment decisions.

In conclusion, our findings demonstrate that thyroid antibody levels significantly influence RI for thyroid hormones, particularly TSH, and that these effects vary by gender and autoanalyzer type. These results highlight the critical need to consider antibody status and demographic factors when establishing RI for thyroid function tests. Moreover, the observed discrepancies between autoanalyzers underscore the urgent need for standardization in thyroid hormone testing. We recommend that future studies establish separate RI for antibody-positive (Ab+) and antibody-negative (Ab−) groups and address existing deficiencies in standardization efforts to improve the accuracy, reliability, and clinical utility of thyroid function assessments.

# Supporting information

**S1 File. Supplementary Tables and Data.** (TSH, fT4, fT3 bias data).
(XLSX)

**S2 File. Pathological Fraction.**
(XLSX)

# Acknowledgments

We thank all participants.

# Author contributions

**Conceptualization:** Muhittin Abdülkadir Serdar, Aysel Ozpinar.

**Data curation:** Havva Beyter, Osman Acar, Oytun Portakal, Özlem Gülbahar, Gülsüm F. Türkeş, Neslihan Yıldırım Saral, Muhittin Abdülkadir Serdar, Aysel Ozpinar.

**Methodology:** Muhittin Abdülkadir Serdar.

**Supervision:** Muhittin Abdülkadir Serdar, Aysel Ozpinar.

**Visualization:** Havva Beyter.

**Writing – original draft:** Havva Beyter.

**Writing – review & editing:** Muhittin Abdülkadir Serdar, Aysel Ozpinar.

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
