## [Decision Letter · Decision Letter 0]

19 Dec 2025

Dear Dr. Ozpinar,

Thank you for submitting your manuscript to PLOS ONE. After careful consideration, we feel that it has merit but does not fully meet PLOS ONE’s publication criteria as it currently stands. Therefore, we invite you to submit a revised version of the manuscript that addresses the points raised during the review process.

We look forward to receiving your revised manuscript.

Kind regards,

Rajeevan Selvaratnam

Academic Editor

PLOS One

3. Please remove all personal information, ensure that the data shared are in accordance with participant consent, and re-upload a fully anonymized data set.

Additional guidance on preparing raw data for publication can be found in our Data Policy (https://journals.plos.org/plosone/s/data-availability#loc-human-research-participant-data-and-other-sensitive-data) and in the following article: http://www.bmj.com/content/340/bmj.c181.long

Additional Editor Comments (if provided):

Reviewers' comments:

Reviewer's Responses to Questions

**Comments to the Author**

1. Is the manuscript technically sound, and do the data support the conclusions?

Reviewer #1: Partly

2. Has the statistical analysis been performed appropriately and rigorously?

Reviewer #1: Yes

3. Have the authors made all data underlying the findings in their manuscript fully available?

Reviewer #1: No

4. Is the manuscript presented in an intelligible fashion and written in standard English?

Reviewer #1: Yes

Reviewer #1: The manuscript entitled “Comparison of Thyroid Hormones Reference Intervals Based on Thyroid Antibody Levels: A multicenter study” aims to determine reference intervals for thyroid hormones based on thyroid antibody levels using different autoanalyzer kits.

This study addresses the fact of being one of the few multicenter studies to establish antibody specific RI for thyroid hormones across four different autoanalyzers.

Regarding the novelty of the article, there are several publications that have previously established RIs for thyroid hormones using the refineR algorithm, some of them have included very large datasets and multiple analyzers as well:

- M.K. Bohn, D. Bailey, C. Balion, G. Cembrowski, C. Collier, V. De Guire, V. Higgins, B. Jung, Z.M. Ali, D. Seccombe, J. Taher, A.K.Y. Tsui, A. Venner, K. Adeli, Reference interval harmonization: harnessing the power of big data analytics to derive common reference intervals across populations and testing platforms, Clin Chem 69 (9) (2023) 991–1008.

-Lewis CW, Raizman JE, Higgins V, Gifford JL, Symonds C, Kline G, Romney J, Doulla M, Huang C, Venner AA. Multidisciplinary approach to redefining thyroid hormone reference intervals with big data analysis. Clin Biochem. 2024 Dec;133-134:110835.

-Helen I.Jansen et al. Age-Specific Reference Intervals for Thyroid-Stimulating Hormones and Free Thyroxine to Optimize Diagnosis of Thyroid Disease. doi: 10.1089/thy.2024.0346.

In regards of the design of the study, it is unclear if thyroid diseases have been properly excluded and could affect the thyroid antibody positive subgroup since the exclusion criteria were based solely on medication treatment and not on thyroid disease. It is recommended to exclude positive thyroid disease patients as exclusion criteria instead of only performing it based on medication treatment.

If available to the researchers, another exclusion criteria that could be considered would be abnormal thyroid ultrasound results.

In regards of thyroid antibodies, were positive anti-thyroid receptor antibodies considered a motive of exclusion?

In regards of exclusion criteria based on medication treatment, did you consider the radioactive iodine (I-131) treatment?

Did you only include primary care patients, or did you include hospitalized patients?

Were results from a single patient excluded from the analysis when multiple results were obtained for that patient within the same year (repeated measurements) to reduce the percentage of pathological results in the datasets?

Did you take into consideration if reflex-testing for assessment of thyroid function could affect your results? A number of laboratories use reflex-testing for assessment of thyroid function, triggering FT4 automatically when TSH results are outside the RI.

I would recommend excluding results from specialists, inpatients, or repeat testing.

Could you provide the pathological fraction from the refineR statistics for each platform?

Being a study performed during several years, please state if significant drift in test result measurement due to reagent reformulation, population drift, seasonal variation, and/or reagent or calibrator lot-to-lot variation/reformulation was taken into consideration.

Please mention if the internal quality control data set was used to ensure the correctness and reliability of the results, and if the laboratories are accredited by the international organization for standardization 15,189 (ISO 15189).

Regarding combination of data from different centers using the same autoanalyzer after the outlier removal, did you perform statistical analysis using the Harris and Boyd method to decide if you could combine data for the estimation of the RIs?

Among the limitations of the study, it has not been mentioned whether the population is iodine sufficient or not, and whether it was taken into account as a variable in the analysis.

Were pre-analytical factors between centers standardized and assessed? Such as the time of collection, time to centrifugation, time to testing, hemolysis, icterus and lipemia indices. If it was not standardized, it should be included as a limitation of the study.

I would also suggest adding in the discussion section a comparison of the results to other publications that have previously defined thyroid hormone RI with refineR, as mentioned previously.

**Do you want your identity to be public for this peer review?** For information about this choice, including consent withdrawal, please see our Privacy Policy

Reviewer #1: No

---

## [Author Response · Author response to Decision Letter 1]

19 Jan 2026

Comments to the Author

1. Is the manuscript technically sound, and do the data support the conclusions?

Reviewer #1: Partly

Şekil

2. Has the statistical analysis been performed appropriately and rigorously?

Reviewer #1: Yes

Şekil

3. Have the authors made all data underlying the findings in their manuscript fully available?

Reviewer #1: No

Şekil

4. Is the manuscript presented in an intelligible fashion and written in standard English?

Reviewer #1: Yes

Şekil

5. Review Comments to the Author

Reviewer #1: The manuscript entitled “Comparison of Thyroid Hormones Reference Intervals Based on Thyroid Antibody Levels: A multicenter study” aims to determine reference intervals for thyroid hormones based on thyroid antibody levels using different autoanalyzer kits.

This study addresses the fact of being one of the few multicenter studies to establish antibody specific RI for thyroid hormones across four different autoanalyzers.

Regarding the novelty of the article, there are several publications that have previously established RIs for thyroid hormones using the refineR algorithm, some of them have included very large datasets and multiple analyzers as well:

- M.K. Bohn, D. Bailey, C. Balion, G. Cembrowski, C. Collier, V. De Guire, V. Higgins, B. Jung, Z.M. Ali, D. Seccombe, J. Taher, A.K.Y. Tsui, A. Venner, K. Adeli, Reference interval harmonization: harnessing the power of big data analytics to derive common reference intervals across populations and testing platforms, Clin Chem 69 (9) (2023) 991–1008.

-Lewis CW, Raizman JE, Higgins V, Gifford JL, Symonds C, Kline G, Romney J, Doulla M, Huang C, Venner AA. Multidisciplinary approach to redefining thyroid hormone reference intervals with big data analysis. Clin Biochem. 2024 Dec;133-134:110835.

-Helen I.Jansen et al. Age-Specific Reference Intervals for Thyroid-Stimulating Hormones and Free Thyroxine to Optimize Diagnosis of Thyroid Disease. doi: 10.1089/thy.2024.0346.

In regards of the design of the study, it is unclear if thyroid diseases have been properly excluded and could affect the thyroid antibody positive subgroup since the exclusion criteria were based solely on medication treatment and not on thyroid disease. It is recommended to exclude positive thyroid disease patients as exclusion criteria instead of only performing it based on medication treatment.

Response: Our exclusion methodology was comprehensive and included both clinical diagnoses and medication use. Specifically, patients with documented thyroid disease diagnoses (based on hospital Laboratory Information System and Hospital Information System records) and/or those receiving thyroid-related medications were systematically excluded from the study. This dual-approach methodology ensures that thyroid disease status was captured regardless of whether patients had formal diagnostic documentation or were being treated with thyroid medications at the time of enrollment. The manuscript text has been revised to clearly articulate this exclusion strategy and to emphasize that thyroid disease screening was not limited solely to medication-based criteria.

If available to the researchers, another exclusion criteria that could be considered would be abnormal thyroid ultrasound results.

Response: In our study, patients for whom thyroid ultrasonography was requested in the Hospital Information System and who were subsequently diagnosed with thyroid disease based on imaging findings were excluded from the analysis.

In regards of thyroid antibodies, were positive anti-thyroid receptor antibodies considered a motive of exclusion?

Response: Thyroid receptor antibody (TRAb) positivity alone was not used as an exclusion criterion in our study. However, patients with a documented diagnosis of Graves' disease, which is characterized by TRAb positivity and associated clinical findings, were systematically excluded from the study.

In regards of exclusion criteria based on medication treatment, did you consider the radioactive iodine (I-131) treatment?

Response: We confirm that our exclusion criteria comprehensively addressed all thyroid-related treatments. Patients who had received any thyroid-related medication or treatment modalities, including radioactive iodine (I-131) therapy, were excluded from the study.

Did you only include primary care patients, or did you include hospitalized patients?

Response: Patients from intensive care units and emergency departments, as well as those whose samples were collected in the afternoon or evening due to the circadian rhythm of TSH, were excluded from the study.

Our study included only primary care outpatients to ensure a representative community-based population. Patients from intensive care units and emergency departments were excluded due to the acute nature of their conditions and potential physiological stress affecting thyroid hormone levels. Additionally, samples collected in the afternoon or evening were excluded from the analysis to minimize the effect of circadian rhythm variations on TSH measurements, as TSH levels demonstrate significant diurnal variation.

Were results from a single patient excluded from the analysis when multiple results were obtained for that patient within the same year (repeated measurements) to reduce the percentage of pathological results in the datasets?

Response: Yes, we carefully addressed the issue of repeated measurements. When multiple samples from the same patient were present in the system within the same year, only the first non-pathological sample was included in the analysis. This approach prevents the overrepresentation of pathological results and ensures that each patient contributes only one measurement to the dataset, thereby maintaining the statistical integrity and reducing potential bias from repeated testing.

Did you take into consideration if reflex-testing for assessment of thyroid function could affect your results? A number of laboratories use reflex-testing for assessment of thyroid function, triggering FT4 automatically when TSH results are outside the RI.

Response: In our country, reflex testing protocols for automatic FT4 measurement when TSH is outside the reference interval are not implemented due to national laboratory regulations and guidelines. Nevertheless, we recognize that in laboratory settings where reflex testing is routinely used, this could influence thyroid function test patterns. This consideration has been added to the limitations section of the manuscript.

I would recommend excluding results from specialists, inpatients, or repeat testing.

Response: We appreciate this recommendation and confirm that our exclusion criteria align with this suggestion. Data from inpatients, including those from intensive care units, endocrinology, nephrology, and emergency departments, were systematically excluded from the study to ensure a representative primary care population. Additionally, as previously detailed, when repeat testing occurred within the same year for individual patients, only the first non-pathological result was retained for analysis. These stringent exclusion criteria were implemented to minimize potential confounding factors and ensure the study population reflects routine primary care laboratory testing patterns rather than specialist or acute care settings. The manuscript has been revised to clearly articulate these exclusion strategies.

Could you provide the pathological fraction from the refineR statistics for each platform?

Response: Yes, as requested, the pathological fractions obtained from the refineR statistics for each platform have been comprehensively added as a supplementary table in the Supplementary Material section.

Being a study performed during several years, please state if significant drift in test result measurement due to reagent reformulation, population drift, seasonal variation, and/or reagent or calibrator lot-to-lot variation/reformulation was taken into consideration.

Response: We acknowledge this is an important methodological consideration for longitudinal studies. Throughout the study period, the same analytical analyzers were consistently used across all systems to maintain standardization. However, we recognize that different reagent lots were applied during the study duration, which may have influenced individual test results.

We believe that accounting for lot-to-lot variation and seasonal variability actually contributes to obtaining more realistic and clinically applicable reference intervals, as these variations reflect real-world laboratory practice. Nevertheless, we acknowledge that reagent reformulations and lot-to-lot variations represent potential sources of systematic variation that could affect reference interval establishment. The potential effects of reagent lot changes, calibrator variations, and seasonal fluctuations have been explicitly added to the limitations section of the manuscript to ensure transparency regarding these confounding factors.

Please mention if the internal quality control data set was used to ensure the correctness and reliability of the results, and if the laboratories are accredited by the international organization for standardization 15,189 (ISO 15189).

Response: One of our hospitals, a private institution, is ISO 15189 accredited. Nevertheless, similar quality requirements (including both internal quality control – IQC and external quality assessment – EQA programs) are mandatory in Turkey and are regularly monitored. During the study period, all laboratories consistently performed the necessary quality control procedures.

We appreciate this important question regarding quality assurance. One of our participating hospitals, a private institution, is ISO 15189 accredited, demonstrating compliance with international standards for medical laboratory competence.

For all participating laboratories, whether ISO 15189 accredited or not, similar rigorous quality requirements are mandatory in Turkey and are regularly monitored by national regulatory authorities. These include both internal quality control (IQC) procedures and external quality assessment (EQA) programs. Throughout the study period, all laboratories consistently performed the necessary quality control procedures according to national guidelines and international best practices. T

Regarding combination of data from different centers using the same autoanalyzer after the outlier removal, did you perform statistical analysis using the Harris and Boyd method to decide if you could combine data for the estimation of the RIs?

Response: No, we did not perform the Harris and Boyd method analysis. We considered this approach inappropriate due to significant differences between the individual systems and the fact that manufacturers recommend different reference intervals for their instruments. However, this method can certainly be performed if requested.

Among the limitations of the study, it has not been mentioned whether the population is iodine sufficient or not, and whether it was taken into account as a variable in the analysis.

Response: Our study did not include direct iodine measurements in the patient population and iodine measurements were not included as a variable in the analysis. Although iodine supplementation is provided through salt fortification and dietary sources in our country, iodine deficiency still exists in the population. (Current iodine status in Turkey. J Endocrinol Invest. 2009 Jul;32(7):617-22. doi: 10.1007/BF03346519. Epub 2009 Jun 24. PMID: 19564718.). We acknowledge this important limitation. This limitation has been added to the limitations section of the manuscript.

Were pre-analytical factors between centers standardized and assessed? Such as the time of collection, time to centrifugation, time to testing, hemolysis, icterus and lipemia indices. If it was not standardized, it should be included as a limitation of the study.

Response: All participating centers regularly perform assessments according to national regulations and apply standardized procedures for pre-analytical phase management. These procedures include standardized protocols for sample collection timing, centrifugation procedures, and time to testing. All participating centers are university hospitals with similar infrastructure and quality standards, which ensures consistency in pre-analytical handling across sites.

I would also suggest adding in the discussion section a comparison of the results to other publications that have previously defined thyroid hormone RI with refineR, as mentioned previously.

Response: The relevant publications defining thyroid hormone reference intervals using the refineR method have been comprehensively added to the discussion section. Our results have been systematically compared with previously published thyroid hormone reference intervals derived using the refineR methodology, highlighting both the similarities and differences in our findings.

---

## [Decision Letter · Decision Letter 1]

9 Feb 2026

Dear Dr. Ozpinar,

Thank you for submitting your manuscript to PLOS ONE. After careful consideration, we feel that it has merit but does not fully meet PLOS ONE’s publication criteria as it currently stands. Therefore, we invite you to submit a revised version of the manuscript that addresses the points raised during the review process.

We look forward to receiving your revised manuscript.

Kind regards,

Rajeevan Selvaratnam

Academic Editor

PLOS One

Journal Requirements:

Reviewers' comments:

Reviewer's Responses to Questions

**Comments to the Author**

Reviewer #1: All comments have been addressed

2. Is the manuscript technically sound, and do the data support the conclusions?

Reviewer #1: Yes

3. Has the statistical analysis been performed appropriately and rigorously?

Reviewer #1: Yes

4. Have the authors made all data underlying the findings in their manuscript fully available?

Reviewer #1: Yes

5. Is the manuscript presented in an intelligible fashion and written in standard English?

Reviewer #1: Yes

Reviewer #1: Firstly, I would like to thank the authors for all the clarifications and adjustments they have made to the manuscript. I only have two minor comments.

Please review these sentences, as they appear to be duplicates:

L152: Moreover, reference intervals for thyroid function tests may vary not only based on antibody 153 status but also due to demographic factors such as ethnicity, age, sex, body mass index, iodine 154 status, and measurement methodology [12-14].

L161: Additionally, RI for thyroid function tests can be affected by 162 demographic characteristics (ethnicity, age, and sex), body mass index, specific medication use, 163 iodine status, and methodological differences

There is a typo in:

L740: FinallyThirdly

**Do you want your identity to be public for this peer review?** For information about this choice, including consent withdrawal, please see our Privacy Policy

Reviewer #1: **Yes:** Alicia Madurga

---

## [Author Response · Author response to Decision Letter 2]

16 Feb 2026

Response to Reviewers

1. If the authors have adequately addressed your comments raised in a previous round of review and you feel that this manuscript is now acceptable for publication, you may indicate that here to bypass the “Comments to the Author” section, enter your conflict of interest statement in the “Confidential to Editor” section, and submit your "Accept" recommendation.

Reviewer #1: All comments have been addressed

2. Is the manuscript technically sound, and do the data support the conclusions?

Reviewer #1: Yes

3. Has the statistical analysis been performed appropriately and rigorously?

Reviewer #1: Yes

4. Have the authors made all data underlying the findings in their manuscript fully available?

Reviewer #1: Yes

5. Is the manuscript presented in an intelligible fashion and written in standard English?

Reviewer #1: Yes

6. Review Comments to the Author

Reviewer #1: Firstly, I would like to thank the authors for all the clarifications and adjustments they have made to the manuscript. I only have two minor comments.

Please review these sentences, as they appear to be duplicates:

L152: Moreover, reference intervals for thyroid function tests may vary not only based on antibody 153 status but also due to demographic factors such as ethnicity, age, sex, body mass index, iodine 154 status, and measurement methodology [12-14].

L161: Additionally, RI for thyroid function tests can be affected by 162 demographic characteristics (ethnicity, age, and sex), body mass index, specific medication use, 163 iodine status, and methodological differences

Response: We have addressed the redundancy by consolidating the information from both sentences into a single, more comprehensive statement. Specifically, we merged the details regarding antibody status, demographic factors, medication use, and methodological differences into one sentence to improve the flow of the paragraph. The redundant sentence at L161 has been removed to improve the clarity and conciseness of the manuscript.

There is a typo in:

L740: FinallyThirdly

Response: The typographical error has been corrected. In the revised manuscript (L746), the term "FinallyThirdly" has been replaced with "Thirdly," to match the preceding "Firstly" and "Secondly" points in the Limitations section.

7. PLOS authors have the option to publish the peer review history of their article (what does this mean?). If published, this will include your full peer review and any attached files.

Do you want your identity to be public for this peer review? For information about this choice, including consent withdrawal, please see our Privacy Policy.

Reviewer #1: Yes: Alicia Madurga

---

## [Editor Report · Decision Letter 2]

18 Feb 2026

Comparison of Thyroid Hormones Reference İntervals Based on Thyroid Antibody Levels: A Multicenter Study

PONE-D-25-52134R2

Dear Dr. Ozpinar,

We’re pleased to inform you that your manuscript has been judged scientifically suitable for publication and will be formally accepted for publication once it meets all outstanding technical requirements.

Kind regards,

Rajeevan Selvaratnam

Academic Editor

PLOS One
---

## [Editor Report · Acceptance letter]

PONE-D-25-52134R2

PLOS One

Dear Dr. Ozpinar,

I'm pleased to inform you that your manuscript has been deemed suitable for publication in PLOS One. Congratulations! Your manuscript is now being handed over to our production team.

Kind regards,

on behalf of

Dr. Rajeevan Selvaratnam

Academic Editor

PLOS One